# Regional Phenotypic Differences of the Opener Muscle in Procambarus *clarkii*: Sarcomere Length, Fiber Diameter, and Force Development

**DOI:** 10.3390/biology9060118

**Published:** 2020-06-05

**Authors:** Rachel C. Holsinger, Robin L. Cooper

**Affiliations:** Department of Biology and Center of Muscle Biology, University of Kentucky, Lexington, KY 40506, USA; rholsinger@sayreschool.org

**Keywords:** invertebrate, neuromodulation, tonic verses phasic fibers, neuromuscular junction

## Abstract

The opener muscle in the walking legs of the crayfish (*Procambarus clarkii*) has three distinct phenotypic regions although innervated by only one excitatory motor neuron. These regions (distal, central, and proximal) have varied biochemistry and physiology, including synaptic structure, troponin-T levels, fiber diameter, input resistance, sarcomere length, and force generation. The force generated by the central fibers when the excitatory neuron was stimulated at 40 Hz was more than the force generated by the other regions. This increase in force was correlated with the central fibers having longer sarcomeres when measured in a relaxed claw. These data support the idea that the central fibers are tonic-like and that the proximal fibers are phasic-like. The addition of serotonin directly on the fibers was hypothesized to increase the force generated by the central fibers more than in the other regions, but this did not occur at 40-Hz stimulation. We hypothesized that the central distal fibers would generate the most force due to the arrangement on the apodeme. This study demonstrates how malleable the motor unit is with modulation and frequency of stimulation.

## 1. Introduction

Arrangement of muscle fibers within a muscle serve particular functional capabilities to optimize the outcome of movements and force development. In addition, the arrangement of sensory structures to monitor the force development, such as Golgi tendon organs, can serve as a means of controlling the amount of force through sensory–CNS–motor unit control, which is assumed to protect the muscle from damage in generating too much force [1,2]. The mechanical arrangement of the Golgi tendon organ in cats has been shown to occur both in series and in parallel with adjacent muscle fibers. In such configuration, the in-series receptor displays a brisk discharge when a muscle is in a stretched position under isometric conditions and a single motor unit is stimulated [3]. This was also shown to occur during passive stretch of muscle [4]. The arrangement of tension receptors in series and in parallel with muscle fibers also occurs in the legs of a crab, Macmillan and Dando [5], and Cooper and Hartman [6] showed that tension receptors are situated on the apodeme in legs of crabs among muscle fiber insertions. The pinnate attachment of the muscle fibers is directly in series with some tension receptors and in parallel with others.

Invertebrate muscles show considerably more variety in their structural organization than vertebrate muscles [7]. Crustaceans do not have tendons, but invaginations of the cuticle referred to as apodemes serve in the same function as tendons. At the distal end of the apodemes in the walking legs are tension receptors that can sense the sheering forces and physiological responses as mammalian Golgi tendon organs. Unlike the muscle bundles in vertebrates, crustacean muscles can have large fibers which attach directly to the apodeme on one end and the cuticle on the other end. Different fiber types (i.e., fast, slow, and intermediate) do exist in different regions of a given muscle in crustaceans; then, some tension receptors may be sensing the force exerted by only one type of muscle fiber over another.

Generally, skeletal muscle can be classified as either slow twitch (tonic) or fast twitch (phasic) based on the speed of contraction and the myosin isoforms causing different ATPase activity. Tonic muscle fibers have longer sarcomeres which contract slower and can generate more force. The opener muscle in the walking legs of crayfish has a pinnate fiber arrangement with three regions: distal, central, and proximal. These different regions have several known characteristics. The proximal region has phasic-like fibers, the central region has tonic-like fibers, and the distal region is a mixed phenotype as determined by the myosin heavy chain isoforms and amount of troponin T [8,9,10]. 

The entire opener muscle and all the fibers are innervated by a single excitatory motor neuron unlike muscles in vertebrates [11]. The analogous opener muscle in the legs of crabs was shown to contain tension receptors only in the distal region of the apodeme [12,13]. Based on this arrangement of fibers and tension receptors, we hypothesized that the distal fibers would generate the most force so that the receptors would be able to limit the amount of force generated by providing an inhibitor feedback onto the opener motor unit.

Serotonin is a neurotransmitter and neuromodulator that is directly released into the hemolymph of crayfish. It can activate readily releasable pools of vesicles in the nerve terminal [14,15]. Since the central fibers are more tonic-like in muscle fiber type it is thought that the nerve terminals have more readily releasable vesicles and that therefore serotonin would have a greater effect on these fibers than on the distal fibers. Thus, we predicted that serotonin would show the strongest response in increasing force generation for the central fibers. Serotonin is known to activate two key second messenger systems, protein kinase C (PKC) and inositol triphosphate (IP_3_), in crustaceans [16,17]; however, there are no investigations in differences in the cellular cascades activated of serotonin on the different muscle fiber types.

The purpose of this project was to correlate the regional differences of the opener muscle with the generation of force, length of sarcomere, and effects of serotonin. Considering that the tension receptors are located in the distal portion of the apodeme, the forces should be able to be relayed to these receptors with respect to the regional differences.

## 2. Methods and Materials

### 2.1. Animals and Care

Medium-sized *P. clarkii* (orbital-thorax lengths are 32 to 44 mm, and the walking leg propodite segments ranged in length from 6.5 to 8 mm and in weight from 0.13 to 0.25 g) were held in individual aquaria of 33 × 28 × 23 cm with a water depth of 10–15 cm. They were fed commercial fish food pellets (Aquadine), and their water was changed weekly. The water was aerated for several days, and carbon-based filters removed the chloramines of local water [18,19] before adding to the aquaria. To make sure that the opener muscle was not compromised, crayfish that appeared to be molting or had visual parasites were excluded from this study. No gravid females were used.

### 2.2. Dissection

The dissection technique permitted a means of force measurement that the intact opener muscle can generate, causing the joint to open with the dactylopodite pushing on the force transducer (see Figure 1).

After inducing the crayfish to autotomize the walking leg by forcibly pinching the ischiopodite segment, a window was cut in the meropodite cuticle to expose the nerve of the opener muscle, and then, a window was cut in the propodite cuticle on the dorsal side.

The preparation was covered with modified Van Harreveld’s solution (in mM: 205 NaCl; 5.3 KCl; 13.5 CaCl_2_.2H_2_O; 2.45 MgCl_2_.6H_2_O; 5 4-(2-hydroxyethyl)-1-piperazineethanesulfonic acid (HEPES) adjusted to pH 7.4 [18]) in a dissecting dish; the cuticle of the meropodite and propodite were removed to expose the nerve bundle and opener muscle, respectively. The nerve bundle was transected to separate the excitatory and inhibitory opener nerves [20]. The excitatory nerve was stimulated using a plastic suction electrode (Figure 2). The preparation was pinned to the recording chamber in a crisscross fashion to make sure that the entire leg did not vibrate when the opener muscle was stimulated. With the dorsal approach, the ventral section of the propodite was pinned to allow the dactylopodite to press against the force transducer pin when the opener muscle was stimulated.

Each leg was weighed, and the dorsal propodite length and orbital-thorax length were measured. At the completion of the experiments, each leg was stained with methylene blue and the opener muscle was photographed at 100× magnification to determine the number of bundles that were ablated.

### 2.3. Stimulation Paradigm

Continual stimulation produced stable force measures and, therefore, was determined to be the best approach. Fifteen seconds at 5, 10, 20, 30, 40, 50, 60, and 70 Hz was used because the force generated leveled off after 10 s at each of the frequencies. When the preparations were stimulated at 80 Hz, depression occurred after 16 s in some preparations and up to 62 s in others. At 40 Hz, the preparation can be stimulated for up to an hour without failure of the nerve.

### 2.4. Measurement of Force

A 2-g range ADInstruments research-grade force transducer (model MLT0402) was used. The transducer was connected to an ADInstruments Bridge Amplifier (ML221), and the support rod of the transducer was attached to a micromanipulator to allow for precise positioning. The transducer was calibrated using 1.00 and 2.00 g weights hung from the attached pin used in the experiments. The force transducer was zeroed using LabChart before calibration and before each experiment.

The dorsal dissection left the entire chelated first walking leg intact, and as the excitatory nerve is stimulated, the claw opens pushing against the force transducer (Figure 3). An “L” shaped pin was attached to the force transducer, which pressed against the dorsal section of the dactylopodite (Figure 4). To make sure that the pin did not slip off the dactylopodite when the claw opened, a small amount of super glue was added to the pin and etched to add a grip. The dactylopodite has a distinctive curved ridge where the pin was repeatedly positioned at the beginning of each experiment. The pin was pressed slightly against the claw to create passive tension between 0.030 g and 0.070 g on the chart recorder. This starting position caused the muscle fibers to be stretched slightly to an almost closed position. Force of the opening claw was recorded using Lab Chart set to an acquisition rate of 2 k/s. The voltage on the stimulator varied depending on each preparation, but the minimum voltage required to open the claw was used.

After the first stimulation paradigm of the intact opener muscle, then muscle bundles corresponding to the proximal, central, and distal regions were systematically cut against the cuticle using a thin curved piece of razor blade (Figure 5). The proximal muscle fibers are closest to the carpopodite and are innervated first by the opener nerve. Both first walking legs were removed from the crayfish used in these experiments, but different tissues (proximal, central, or distal) were ablated from each leg. First, only the distal fibers were detached from the cuticle and the force that the opening claw could generate with the stimulation paradigm was recorded (Figure 5B). The preparations were exposed to serotonin (400 nM), and after 30 s of incubation, the stimulation paradigms were repeated.

Then, in another five preparations, only the central fibers were ablated (Figure 5C). The same methods were used for an additional five preparations where only the proximal fibers were ablated. When the proximal fibers were cut, no force could be measured by the distal and central ones alone; therefore, to try to determine the contribution of the proximal fibers, the central and distal fibers were cut at the same time in another 5 preparations (Figure 5D). Next, in five preparations, the distal fibers were ablated, tension was recorded, and then the central fibers of the same preparation were cut and tension was recorded. Serotonin was added only after the fibers in both regions were removed from the cuticle. Finally, five controls were recorded; the paradigms 5, 10, 20, 30, 40, 50, 60, and 70 Hz for 15 s each were completed twice; and the force was measured before exposing the preparations to serotonin (400 nM).

### 2.5. Electrophysiology

The excitatory axon innervating the opener muscle in the crayfish was stimulated in the meropodite by placing a branch of the leg nerve into a suction electrode connected to a Grass stimulator [20]. Stimulation at 40-Hz trains of 30 pulses in duration was applied to the excitatory nerve to compare the responses obtained in the three general regions of the opener muscle: distal, central, and proximal). Preparations were used immediately after dissection, and all the experiments were performed at room temperatures (19–21 °C).

### 2.6. Muscle Anatomy

After electrophysiological measures, the opener muscle fibers were fully stretched in a natural position by closing the chelated ends. The fibers were fixed with Bouin’s fixative, and a few bundles of fibers from the three regions were removed and mounted on glass slides. The sarcomere lengths were measured five times in each of the five preparations in the three different regions at 100×, and an average was taken.

### 2.7. Statistical Analysis

The force values being analyzed always started at 0.000 g after subtracting any initial force on the transduced from positioning the pin. Although three independent variables were tested (stimulation, ablated tissue, and addition of serotonin), the data did not meet the assumptions for a multivariate analysis of variance (MANOVA) due to the small sample size and non-normal distribution of data. Thus, nonparametric statistics, Mann–Whitney test, Kruskal–Wallis one-way analysis of variance by ranks, and Wilcoxon Matched-pairs signed-ranks test were used.

The experiments were designed to determine if there were a significant difference in the force generated between the control legs, ablated distal, ablated central, and ablated distal/central at 40 Hz. Since the probability of finding an invalid significant result increases as the number of comparisons increase, only the force generated from 40–50 Hz was used to reduce the number of statistical tests performed and the chance of erroneous statistical results.

Force development at 40 Hz was used for statistical analysis because all intact preparations showed increase in force above ±0.003 error of the force transducer at 30 Hz and 40 Hz. Stimulation frequencies below 30 Hz did not cause force generation above noise in all intact preparations, with 5 Hz only having 11% of preparations generating measurable force. Above 40 Hz, the force generated reached the upper limit of the transducer on 11 of the 35 preparations and higher stimulation caused depression in 5 of the 35 preparations. Force was maintained at 40 Hz for at least five minutes in this study to examine if depression in force occurred, and excitatory junction potentials (EJPs) have been measured for well over an hour in other studies [18]. The variability in force generation in the different crayfish cannot be controlled so a percent difference was used to compare within the data set.

## 3. Results

### 3.1. Force Generation in Three Regions of Opener Muscle

Force was generated by stimulating the motor neuron at 5, 10, 20, 30, 40, 50, 60, and 70 Hz for 15 s each. The force was measured at the plateau of each stimulation. The forces with intact and ablated reginal fibers are shown for representative preparations in Figure 6. The forces for the intact muscle and then with the distal fibers ablated, followed next by the ablation of the central fibers, reveals the contribution of the distal and central fibers to the intact muscle (Figure 6A). In a different preparation in which the force for the intact fibers is measured followed by ablation of the central fibers (Figure 6B) illustrates the contribution of the distal and proximal fibers in comparison to the total force of the intact preparation.

In the first set of experiments, the increase in force generated by the intact opener muscle was conducted. The mean (+/− SEM) force for 34 preparations was conducted prior to ablating the regions of muscle fibers (Figure 7A). In a subset of these preparations, the distal fibers were ablated and the forces were re-measured with the same stimulation paradigm (Figure 7B). In another subset of the original preparations, the central fibers were ablated and forces were measured again with the stimulation paradigm (Figure 7C). In the last subset of the starting preparations, both the distal and central fibers were ablated, leaving only the proximal fibers to generate force (Figure 7D).

In order to examine the force generated by the proximal fibers the distal fibers and then central fibers were ablated (Figure 7E). As with ablating the distal and central fibers at the same time (Figure 7), the proximal fibers have an increase in force generation starting at 50 to 60 Hz stimulation frequency. As expected, there was no significant difference in force generated at 40 Hz between the two stimulations of control preparations by the Wilcoxon Matched Pairs signed rank test (Figure 8, z = 0.813, *p* < 0.05). There is a significant difference between all other preparations except between the control and ablated distal preparations. There was a significant difference in the force that was generated at 40 Hz between the control and ablated central (H = 12.071 *p* < 0.05, Kruskal–Wallis), the control and all 13 ablated central/distal (H = 9.929 *p* < 0.05, Kruskal–Wallis test), the ablated central and ablated distal (H = 10.929, *p* < 0.05, Kruskal–Wallis test), and the distal to all ablated central/distal (H = 8.786, *p* < 0.05, Kruskal–Wallis test). However, no significance was found between either ablated central to ablated distal/central (H = 2.143) nor between the control and ablated distal (H = 1.143).

Because the proximal fibers could not be cut without causing major damage to the nerve, direct measurement of the force generated by the distal and central regions alone is impossible. However, by subtracting different measurements, one could examine the contributions of the central fibers and distal fibers (Table 1). Therefore, one can conclude that the distal fibers contribute the least to the force generation at 40 Hz (0.004 g) and that the central fibers contribute the most (0.058 g). The proximal fibers are necessary for proper alignment of the muscle on the apodeme and contribute more to the total force generation than the distal but less than the central (0.015 g).

To determine the amount of force the central fibers contribute, the force generated from the ablated distal/central fibers was subtracted from the ablated distal. Then to determine the force the distal fibers contribute, the force from the force of the ablated distal/central fibers was subtracted from the force of the ablated central. Table 1 shows these values.

### 3.2. Sarcomere Length in Three Regions of Opener Muscle

Prior research has reported that the crayfish phasic fibers have sarcomere lengths of 2–3 µm, and tonic fibers are 6–12 µm. Figure 9 shows representative sarcomere size differences in the different regions of the crayfish opener muscle. Figure 10 shows that the central fibers have longer sarcomeres than the distal or proximal. Significant differences were present for the distal and proximal compared to the central but not between distal and proximal (Holm–Sidak multiple comparison, *p* < 0.05).

The sarcomeres in the central region are longest when passively stretched supporting muscle fibers in this region are more tonic-like. The sarcomeres in the proximal region are the shortest, supporting muscle fibers in this region are more phasic-like, and the sarcomeres in the distal region are in between the proximal and central lengths.

### 3.3. Effect of Serotonin on Force Generation

After stimulating the opener muscle twice with the same paradigm, serotonin was added before stimulating the muscle again. The same procedures were repeated for ablating different regions as mentioned previously. The distal only was ablated with serotonin added to examine the effect of the central and proximal groups (Figure 11B), and then, only the central fibers were ablated, and serotonin was added (Figure 11C). Lastly, distal and central fibers were ablated concurrently, leaving only the proximal fibers (Figure 11D). When both the central and distal fibers were ablated together and only the proximal remaining no differences were observed from ablating one region at a time (Figure 11E).

The change in force from before and after the exposure to serotonin was determined at 40 Hz as an index to the effect of serotonin on force development (Figure 12). There was a significant difference between the amount of force generated at 40 Hz between control and added serotonin (Wilcoxon Matched Pairs signed ranks, *p* = 0.047), ablated central and added serotonin (Wilcoxon Matched Pairs signed ranks, *p* = 0.05), and ablated central/distal and added serotonin (Wilcoxon Matched Pairs signed ranks, *p* = 0.027). There is no significant difference between ablated distal and added serotonin (Wilcoxon Matched Pairs signed ranks, *p* = 0.219).

It was expected that central fibers were most affected by addition of serotonin; however, this was not the case (Table 2). The force generated by the proximal fibers was subtracted from the ablated distal and ablated central for the 40 Hz stimulation paradigm. The distal fibers were most affected by the addition of the serotonin and generated the most force (0.015 g); then, the proximal (0.009 g) and the central (0.006 g) were the least affected.

## 4. Discussion

The data in this investigation indicated that the distal fibers contribute the least to the force generation, that the central fibers contribute the most, but that the proximal fibers are necessary for proper alignment of the muscle on the apodeme and contribute more to the total force generation than the distal but less than the central. There was no significant difference between the control and ablated distal preparations. This lack of difference in force generation when the distal fibers were ablated could show that these fibers contribute a small amount of the total force. It is interesting that the distal fibers appear to contribute the least to the total force generated because the tension receptor neurons that sense force development of the muscle are located in the distal region of the apodeme. The force generation and sarcomere length supported the notion that central fibers are tonic-like fibers and that the proximal fibers are phasic like by generating the least force with the shortest sarcomeres.

The high-output proximal regions have a larger enhancement of excitatory junction potential (EJP) amplitude and larger facilitation index when compared to the low-output central region [18]. The central region contributed the most to the total force generation, but it would be interesting to determine if the slope of force generation is different for the proximal and central fibers. It would be expected that the proximal region would generate force more quickly because of the large amplitude of the initial EJP but this force would not be maintained and therefore would not contribute as much to the overall force generated. The fiber diameter and input resistance correlate for the three regions. The larger diameter fibers have a lower input resistance. These larger fibers have a greater surface area, and if one assumes the leak channels of the membrane are uniform, then one would expect the larger surface area to have a lower input resistance [21]. The specific membrane resistance can be estimated based on the input resistance and specific internal resistance of the muscle fiber [22,23]. The input resistance alone does not account for the large differences in EJP amplitudes between the proximal fibers and the central or distal fibers. Thus, the differences in input resistance of the muscle fibers alone cannot account for the entire EJP amplitude differences. The difference in synaptic efficacy with respect to mean quantal content on the different muscle fibers likely accounts for the majority of the differences [11,24].

Serotonin is thought to enhance synaptic transmission independent of Ca^2+^ levels in the cell. Serotonin caused an increase in the frequency of spontaneous events and therefore increases probability of neurotransmitter release and increased EJP amplitude [21]. Glusman and Kravitz were able to show that the number of failures decrease when serotonin is added [25]. The mechanism of serotonin effect when applied directly on muscle might be related to the activation of two different second messenger systems: activation of protein kinase C (PKC) and production of inositol triphosphate (IP_3_) [16,17]. This action could increase the input resistance and account for some of the increase in EJP amplitude. The serotonin receptor subtypes in crayfish opener muscle is suggested to be a 5-HT2 receptor subtype based on pharmacological actions [26]. Molecular sequencing of isolated neural tissue from crayfish indicated that 5-HT_2β_ and 5-HT_1α_ are present [27]. It would be of interest to know if regional differences in 5-HT receptors subtypes and density exist across the muscle fibers of the opener muscle and presynaptic nerve terminals. The entire muscle is innervated by a single excitatory motor neuron, but differences in modulation on the distant muscle regions may be different. There are no reports addressing if there are differences in the cellular action of serotonin on the different muscle fiber types. A possible mechanism for serotonin to increase force generation in the opener muscle is through activation of G coupled receptors and the 1,4,5-trisphosphate (IP3) and diacylglycerol (DAG) pathway. The production of IP3 can directly result in Ca^2+^ release from the sarcoplasmic reticulum through IP3 receptors on the sarcoplasmic reticulum [28]. It has also been shown in *Aplysia* (sea slug) that serotonin could cause the phosphorylation of synapsins, thus increasing the likelihood that vesicles would become untethered from the cytoskeleton and dock with the presynaptic membrane [29]. This could account for the increased occurrence of spontaneous quantal events and enhanced evoked responses with serotonin exposure [30].

Future studies would be helpful to examine individual fibers from the different regions in the degree of force able to be generated to correlate to the length and width of the fiber. Then, perhaps a computation model with the angle of pinnation and force exerted on the apodeme for the fibers could be theoretically determined to see if they relatively match the results presented herein.

## 5. Conclusions

The opener muscle in crayfish has three distinct phenotypic regions (distal, central, and proximal). The central fibers contribute the most for the force generated in opening the dactylopodite upon nerve stimulation. The central muscle fiber has a slow phenotype with longer sarcomeres than the other two regions. The motor units in each region are modulated by serotonin which enhances the amount of force generated.

## Figures and Tables

**Figure 1 biology-09-00118-f001:**
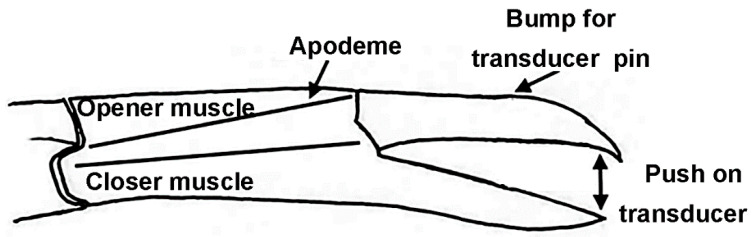
Side view of the chelated first walking leg. pushing on the transducer pin resting on the top of the dactylopodite. This side view shows the location of the apodemes for opener and closer muscles in the propodite. The force transducer pin rests on the bump on the dactylopodite.

**Figure 2 biology-09-00118-f002:**
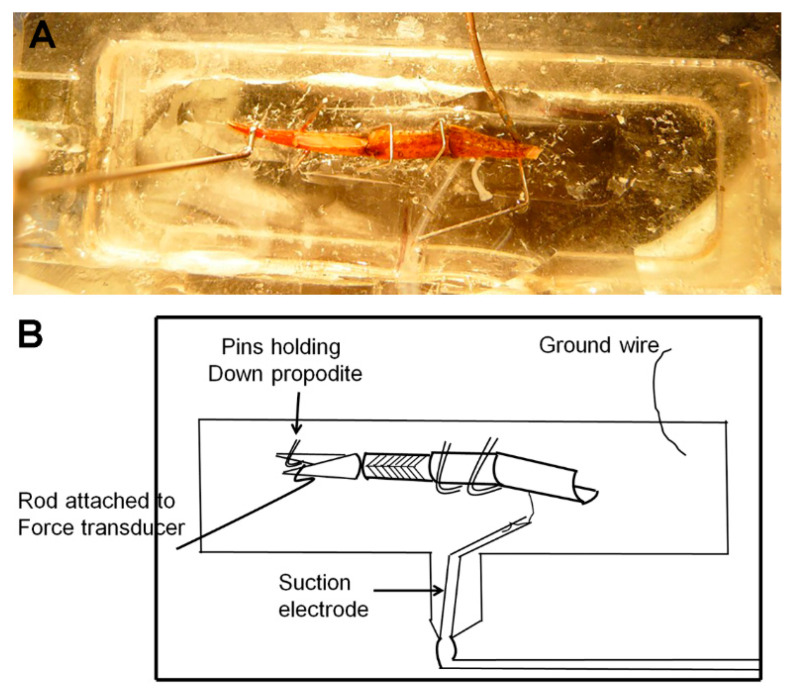
Recording chamber showing pinning of preparation and suction electrode: (**A**) The photograph shows a dorsally dissected preparation pinned in the recording dish with the force transducer pin resting on the dactylopodite. The suction electrode holding the excitatory nerve can be seen in the bottom center of the dish, and the inhibitory nerve is floating free. (**B**) A diagram detailing the recording chamber setup specifically showing how the preparation is pinned between the dactylopodite and ventral propodite.

**Figure 3 biology-09-00118-f003:**
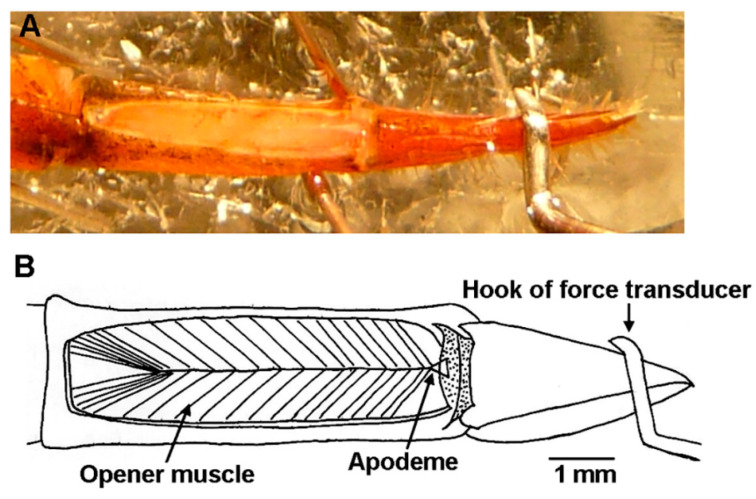
Dorsal dissection of first walking leg with force transducer location: (**A**) Photograph of the first walking leg showing location of the force transducer hook resting on the dactylopodite. (**B**) Diagram detailing anatomy of the preparation showing the pinnate arrangement of the opener muscle and its attachment to the apodeme.

**Figure 4 biology-09-00118-f004:**
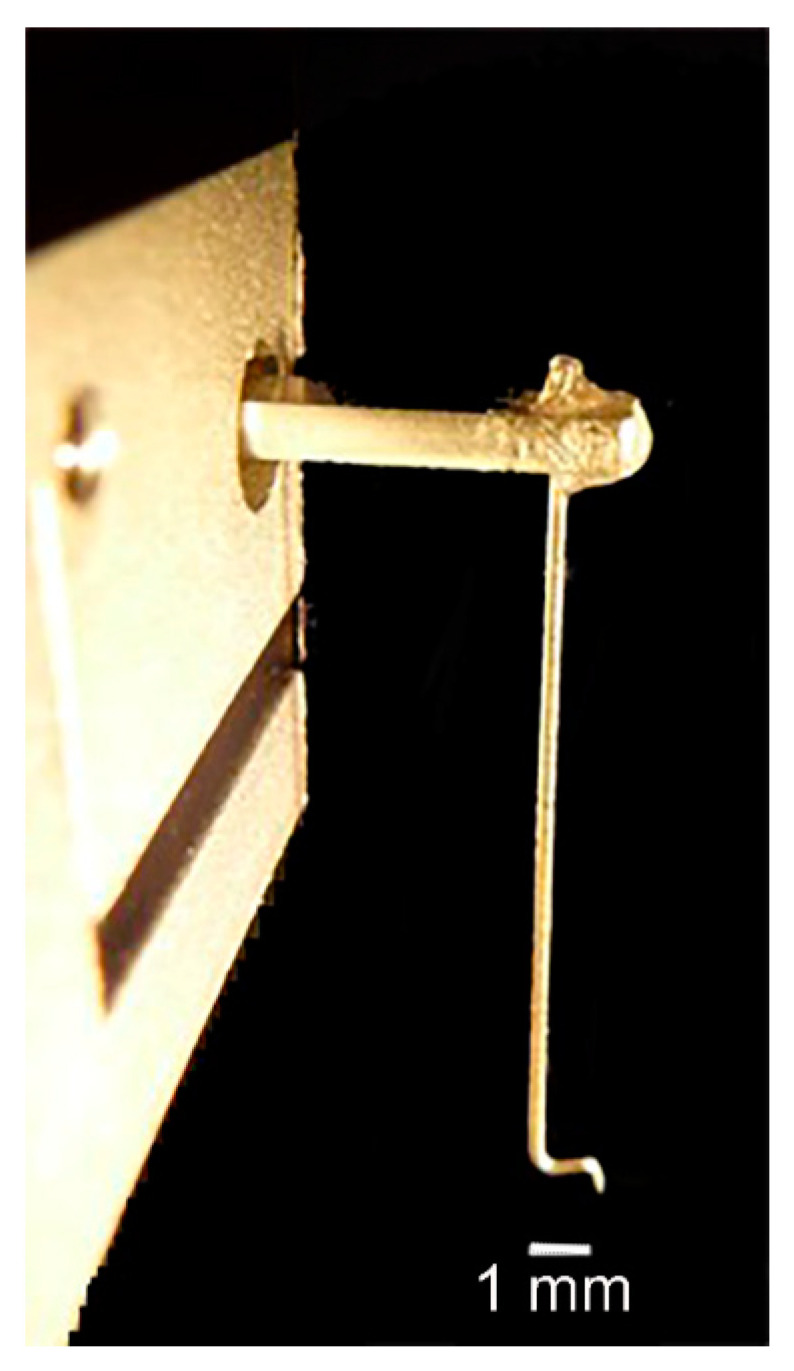
Force transducer “L”-shaped pin design: This pin rests on the bump on the dactylopodite and relays the force to the transducer as the cheliped opens with stimulation.

**Figure 5 biology-09-00118-f005:**
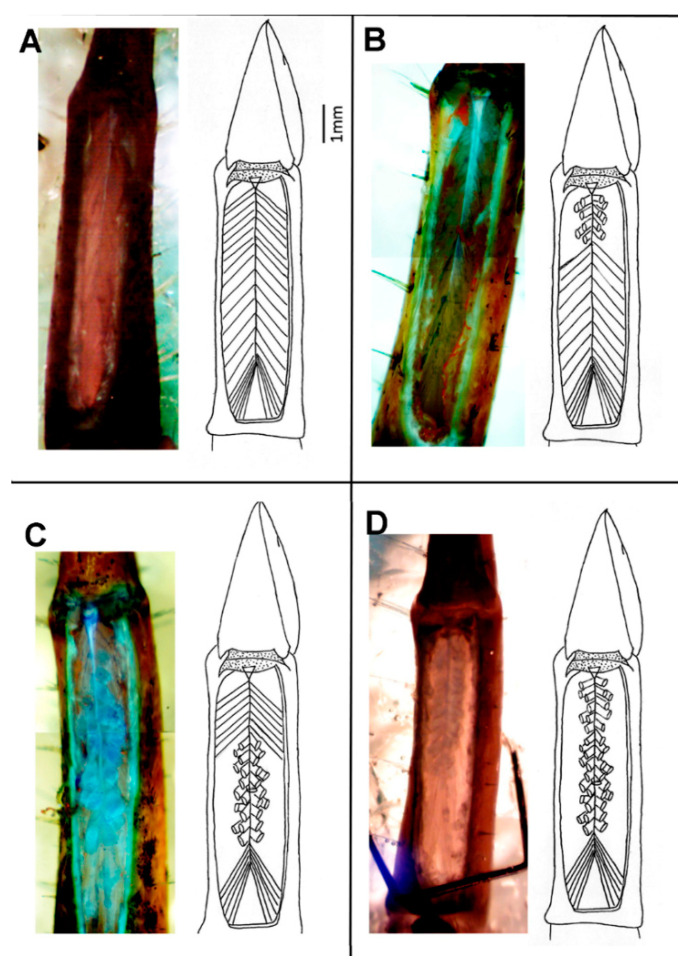
Dorsal dissections of the first walking leg showing ablated tissue: The photographs show the ablated fibers are stained more darkly by methylene blue than the intact fibers. Four different dissections were performed: (**A**) intact opener muscle control, (**B**) ablated distal fibers, (**C**) ablated central fibers, and (**D**) ablated distal and central fibers.

**Figure 6 biology-09-00118-f006:**
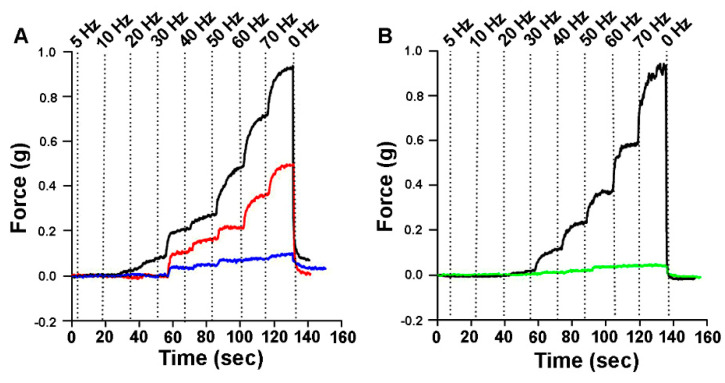
Forces of intact before and after ablated regions of fibers of the opener muscle over various stimulation frequencies: (**A**) The force generated by one preparation stimulated over 145 s: the top trace shows force generated by the intact opener muscle, the middle trace is the force generated with the distal fibers ablated, and the bottom trace is the force generated by the same preparation when the central fibers were ablated in addition to the distal fibers. (**B**) The force generated in a different preparation: The top trace shows force generated by the intact opener muscle, and the lower trace is the force generated with central fibers ablated. The vertical dotted lines indicate the start of the stimulation frequency listed at the top.

**Figure 7 biology-09-00118-f007:**
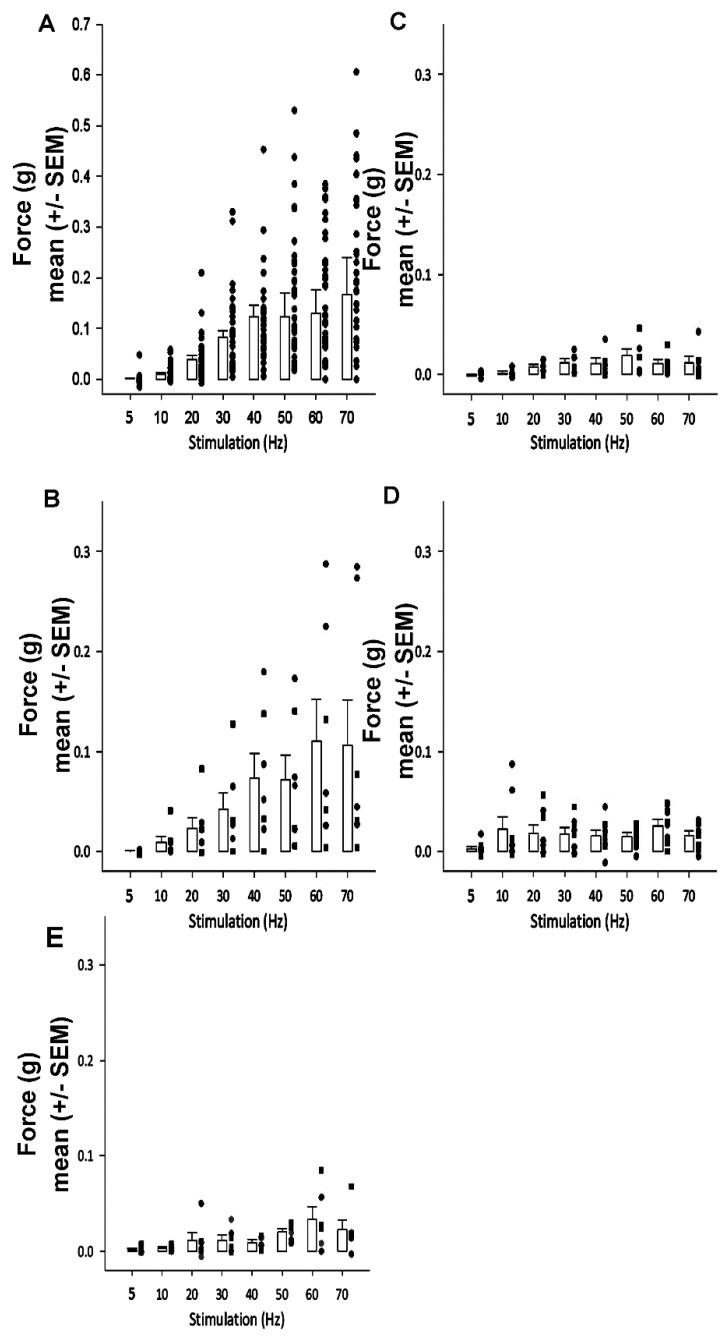
Change in force over stimulation paradigm with before and after regions of fibers being ablated: The bar graphs are of the mean and standard error of the mean (+/− SEM) at each of the 8 stimulation frequencies. Each dot represents a preparation. (**A**) The force generated by the intact opener muscle for all 34 preparations prior to ablating the tissue. (**B**) The force generated by the central and proximal fibers when the distal fibers were ablated. (**C**) The force generated by the distal and proximal fibers when the central fibers were ablated. (**D**) The force that can be generated by the proximal fibers alone when both central and distal fibers were ablated at the same time. (**E**) The force that can be generated by the proximal fibers when the central and proximal fibers were ablated.

**Figure 8 biology-09-00118-f008:**
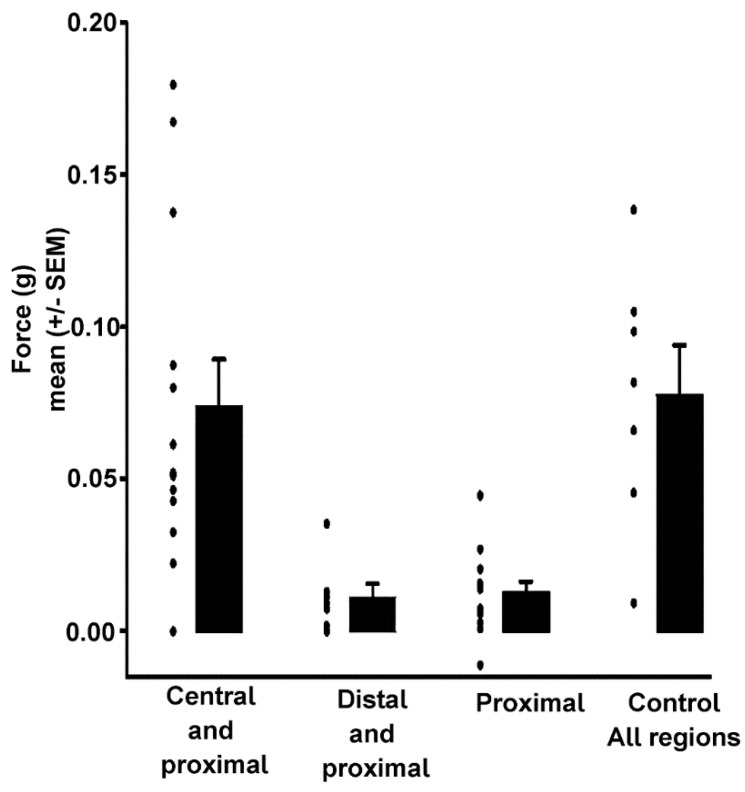
Force development at 40 Hz: The amount of force that can be generated by the central and proximal tissues together is more than the amount the proximal alone can generate. When the central fibers are ablated and similar amount of muscle fibers remain as in the ablated distal preparations, there is a significant decrease in the amount of force that the muscle can generate. There is no significant difference in the control preparations and the ablated distal preparations because the distal do not appear to be responsible for much of the total force generation.

**Figure 9 biology-09-00118-f009:**
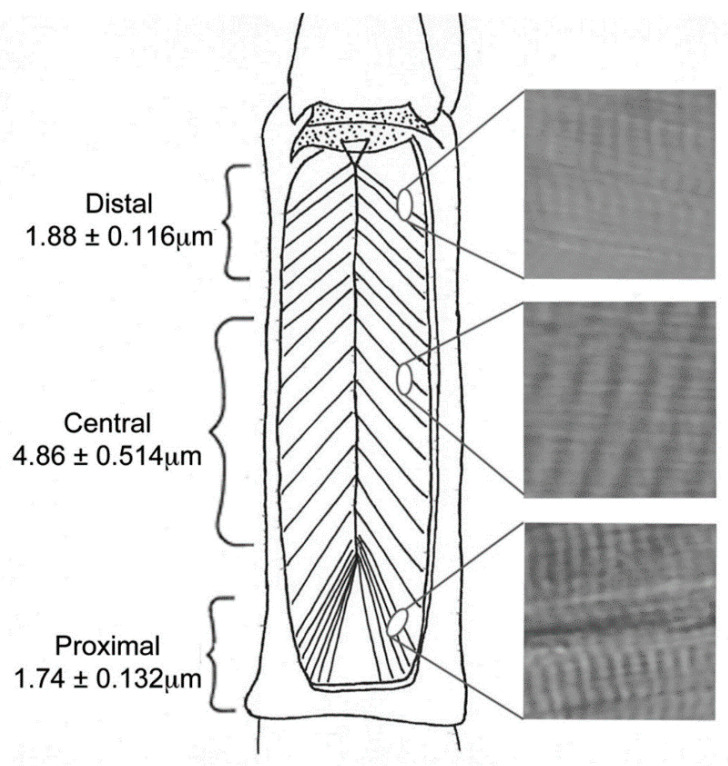
Sarcomere lengths: The sarcomeres of each of the three regions were passively stretched by pushing the claws of 5 different legs closed. Each region was measured 5 times in each preparation for a total of 25 measurements. Highlighted to the right of this figure are sample photographs. The central region had larger sarcomeres than the proximal or distal.

**Figure 10 biology-09-00118-f010:**
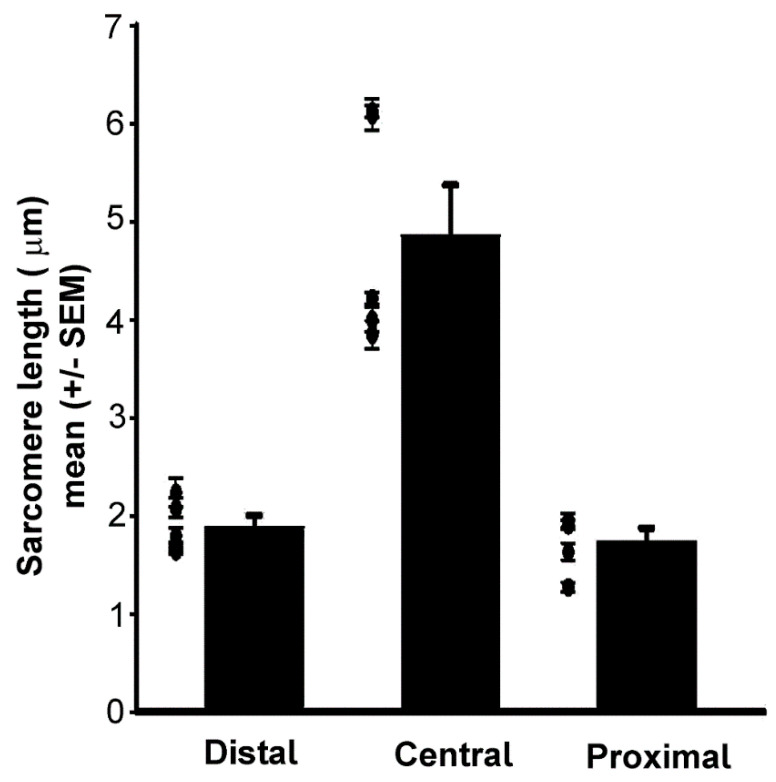
Regional differences in sarcomere lengths: Each dot represents the average measurement in each of the 5 preparations. The bar represents the average of those preparations (+/− SEM). The sarcomeres in the central region are larger than those in the proximal and distal. Significant differences for the distal and proximal compared to the central but not between distal and proximal (Holm–Sidak multiple comparison, *p* < 0.05).

**Figure 11 biology-09-00118-f011:**
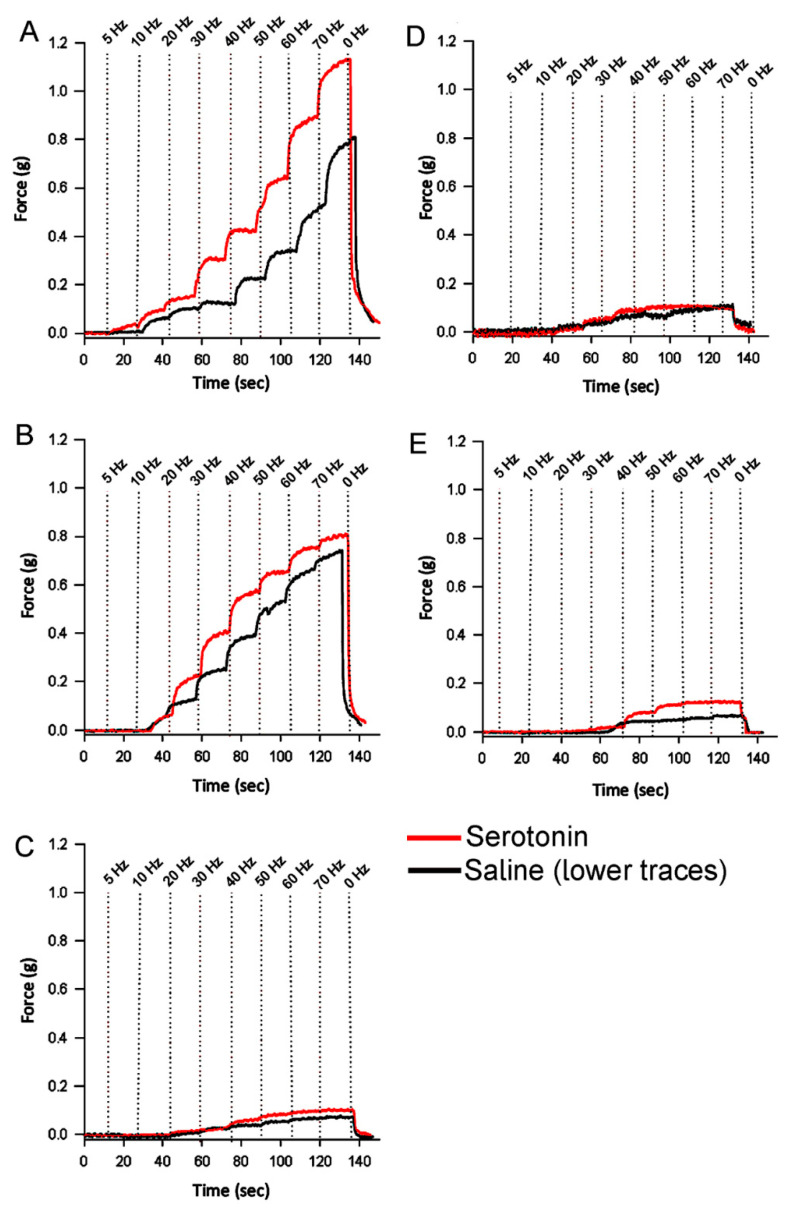
Representative traces of force generation before and after exposure to serotonin: In each graph, the black trace (lower trace) represents the force generated by fibers without serotonin and the red trace (upper trace) represents the force generated in the presence of serotonin. Serotonin increased the force generated in each of the different preparations: (**A**) the control, (**B**) ablated distal fibers, (**C**) ablated central fibers, (**D**) concurrently ablated distal and central fibers, and (**E**) ablated distal then central fibers.

**Figure 12 biology-09-00118-f012:**
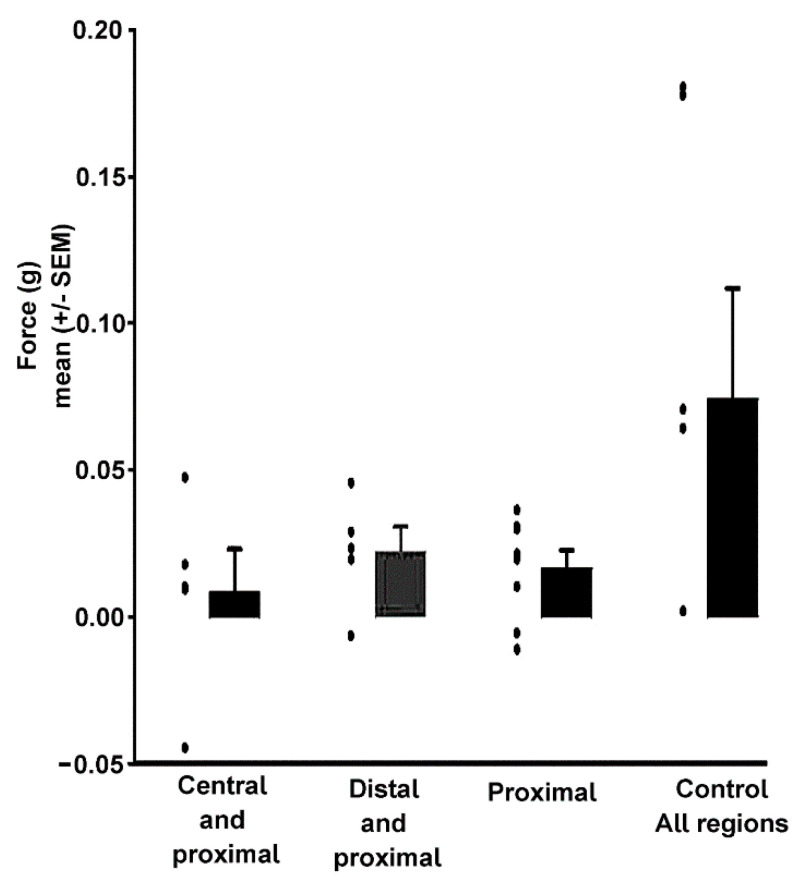
Effect of serotonin on force development for stimulation frequency at 40 Hz: This is the additional force (not total force) that was generated by the fibers during serotonin exposure. The mean (+/−SEM) are presented for the groups along with the values for each individual preparation as a dot.

**Table 1 biology-09-00118-t001:** Amount of calculated force (grams) generated by the intact tissue.

Frequency of Stimulation								
Intact Tissue	5 Hz	10 Hz	20 Hz	30 Hz	40 Hz	50 Hz	60 Hz	70 Hz
**Distal**	0.003	0.021	0.013	0.008	0.004	0.005	0.011	0.015
**Central**	0.003	0.014	0.004	0.024	0.058	0.058	0.085	0.091
**Proximal**	0.003	0.022	0.019	0.018	0.015	0.015	0.026	0.016

The average amount of force generated by 5 or more preparations.

**Table 2 biology-09-00118-t002:** Increase in calculated force (grams) generated by the intact tissue after serotonin was added

Frequency of Stimulation								
Intact Tissue	5 Hz	10 Hz	20 Hz	30 Hz	40 Hz	50 Hz	60 Hz	70 Hz
Distal	0.001	0.0216	−0.013	0.0167	0.015	0.005	0.048	0.012
Central	0.001	0.011	0.033	0.061	0.006	0.02	0.031	0.049
Proximal	0.002	−0.017	0.013	−0.005	0.009	0.013	−0.013	−0.009

The average amount of force generated by 5 or more preparations without serotonin was subtracted from the forces with serotonin.

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
