# Peer review of "Regional Phenotypic Differences of the Opener Muscle in Procambarus clarkii: Sarcomere Length, Fiber Diameter, and Force Development"

_biology, 2020, doi:10.3390/biology9060118_

Round 1

Reviewer 1 Report

I consider the manuscript of Holsinger and Cooper of interest for scientific community.

Abstract and introduction are accurate and informative regarding the objectives. The data are well organized and well described. Finally, the conclusion is consistent with the discussion and results sections.

I have only some minor suggestions and concerns listed below.

-       I suggest to add information about the concentration of serotonin used to study the effect of serotonin on force generation and stimulation of the opener muscle.

-       Moreover, I suggest to discuss the paper of Spitzer and coworkers on serotonin receptors identified in crayfish “Conservation of structure, signaling and pharmacology between two serotonin receptor subtypes from decapod crustaceans, Panulirus interruptus and Procambarus clarkia. J Exp Biol. 2008 January ; 211(0 1): 92–105. doi:10.1242/jeb.012450.”

I think that important future analysis could be directed in order to identify the possible mechanism of action of serotonin by using for example drugs specific to 5-HT receptor subtypes in this opener neuromuscular preparation of crustaceans.

Reviewer 2 Report

This paper on crustacean muscle force generation is technically sound, but lengthy and would greatly benefit from revision.

Major issues:

The Methods section repeatedly switches tenses. It should use past tense throughout.

The presentation of statistics is frequently inadequate or confusing.

  • Figures 13, 16: No indication of what error bars show. (SD, SE, CI?)
  • Lines 351-353: p values with no test statistic or degrees of freedom, and lack of clarity to test (presumably Wilcoxen test).
  • Line 310, Figure 13: Nonstandard use of "T" to report chi-squared test.
  • Line 368-369: Unitless numbers are completely ambiguous.

Minor issues:

Occasional use of singular first person ("I") in a paper with two authors.

The Introduction does not lead to the project. The project is about force generation, but the Introduction instead details how crustacean and vertebrate muscles differ, which is not highly relevant here.

The Methods section contains diversions into rejected protocols, which add no value for the reader.

Information needed to interpret figures is often in the main text instead of the figure legend.

Many figures would be easier to interpret with a legend.

Many figures that show the preparations are low quality and "sketchy," and should be cleaned up in a graphics editor. In particular, Figure 2B has many wavy lines that should be straight.

Some figures have components that are not described. E.g., stimulus rates are shown as vertical dotted lines in some graphs, but these never explicitly mentioned.

Figures 10, 13 have colours that serve no purpose.

The Graphic Abstract shows no results, and so does not serve the purpose of an abstract.

PDF with more comments attached.
